# Pharmacokinetic Interpretation of Applying Local Drug Delivery System for the Treatment of Deep Surgical Site Infection in the Spine

**DOI:** 10.3390/pharmaceutics16010094

**Published:** 2024-01-10

**Authors:** Ahmad Khalid Madadi, Moon-Jun Sohn

**Affiliations:** 1Department of Biomedical Science, Graduate School of Medicine, Inje University, 75, Bokji-ro, Busanjingu, Busan 47392, Republic of Korea; khalidmadadi@yahoo.com; 2Department of Neurosurgery, Neuroscience & Radiosurgery Hybrid Research Center, College of Medicine, Inje University Ilsan Paik Hospital, Juhwa-ro 170, Ilsanseo-gu, Goyang City 10380, Republic of Korea

**Keywords:** surgical site infection, drug delivery system, polymethylmehthacrylate (PMMA), drug release kinetic, sustained release, antibiotics-loaded bone cement, mathematical model

## Abstract

Surgical site infections (SSIs) after spinal surgery present significant challenges, including poor antibiotic penetration and biofilm formation on implants, leading to frequent treatment failures. Polymethylmethacrylate (PMMA) is widely used for localized drug delivery in bone infections, yet quantifying individual drug release kinetics is often impractical. This retrospective study analyzed 23 cases of deep SSIs (DSSIs) following spinal surgery treated with antibiotic-loaded PMMA. A mathematical model estimated personalized drug release kinetics from PMMA, considering disease types, pathogens, and various antibiotics. The study found that vancomycin (VAN), ceftriaxone (CRO), and ceftazidime (CAZ) reached peak concentrations of 15.43%, 15.42%, and 15.41%, respectively, within the first two days, which was followed by a lag phase (4.91–4.92%) on days 2–3. On days 5–7, concentrations stabilized, with CRO at 3.22% and CAZ/VAN between 3.63% and 3.65%, averaging 75.4 µg/cm^2^. Key factors influencing release kinetics include solubility, diffusivity, porosity, tortuosity, and bead diameter. Notably, a patient with a low glomerular filtration rate (ASA IV) was successfully treated with a shortened 9-day intravenous VAN regimen, avoiding systemic complications. This study affirms the effectiveness of local drug delivery systems (DDS) in treating DSSIs and underscores the value of mathematical modeling in determining drug release kinetics. Further research is essential to optimize release rates and durations and to mitigate risks of burst release and tissue toxicity.

## 1. Introduction

Surgical site infections (SSIs) are prevalent and serious healthcare-associated infections, posing risks of patient mortality and increased healthcare costs. The rates of readmissions due to SSIs vary based on the population, type of surgery, and instrument usage [1]. Common pathogens responsible for SSIs include methicillin-sensitive/resistant *Staphylococcus aureus* (MSSA/MRSA), coagulase-negative staphylococci (CoNS), *Enterococcus* sp., and *Escherichia coli* [2]. A meta-analysis encompassing 27 studies and 22,475 patients reported an overall SSI rate of 3.1% with superficial SSIs at 1.4% and deep SSIs (DSSIs) at 1.7% [3]. SSIs following spinal surgery specifically affect intervertebral discs and surrounding soft and articular tissues [4].

Managing DSSIs, especially in cases involving surgical instrumentation, requires a multifaceted approach. This includes improved antibiotic penetration, addressing antibiotic resistance, reducing recurrence, and implementing effective local drug delivery systems (DDS). Standard treatment often involves surgical debridement and extended antibiotic therapy. Alternative strategies, such as local vancomycin (VAN) powder application, negative pressure wound therapy, and continuous saline irrigation, are being explored. Yet, there remains no consensus on the optimal treatment for SSIs post-instrumented spinal surgery [5]. Key challenges in SSI management include antibiotic selection, overcoming resistance, and addressing limitations in combating biofilms or intracellular forms [4].

Local antibiotic delivery systems, while promising, face challenges such as antibiotic choice, carrier compatibility, systemic absorption, timing, duration, resistance development, and the need for carrier removal or exchange. Specialized carriers in local DDS offer benefits like improved penetration, sustained release, and localized concentrations, minimizing systemic side effects [6]. Antibiotics can be delivered with or without a carrier; various carriers include polymethylmethacrylate (PMMA) bone cement, calcium sulfate, calcium hydroxyapatite, bioactive glass, and hydrogels being used [7]. This approach achieves high local antibiotic concentrations, exceeding the minimum inhibitory concentration (MIC) while reducing systemic levels and associated toxicity, effectively combating biofilms. Combining debridement with short-term antibiotic-impregnated cement placement has shown improved outcomes compared to surgery alone. Moreover, integrating local antibiotic delivery with systemic therapy has demonstrated superiority over systemic therapy alone in two-stage arthroplasty revision procedures [8]. PMMA is widely utilized in local DDS but presents several drawbacks, including the necessity for removal surgery, the potential for prolonged sub-inhibitory antibiotic release leading to resistance, and unsuitability for heat-sensitive antibiotics. Despite these limitations, local antibiotic delivery via PMMA maintains high local concentrations while minimizing systemic toxicity, often resulting in enhanced outcomes when combined with systemic therapy [8].

A significant challenge in using PMMA for sustainable drug release in DSSIs is the scarcity of clinical data. Local antibiotic delivery is promising for managing infections post-spinal surgery, yet further research is needed to optimize these systems and fully understand their benefits and limitations. The pharmacokinetic approach to local DDS with PMMA involves impregnating antibiotics into PMMA carriers, allowing for their gradual release, which provides sustained and localized concentrations at the infection site. The release kinetics of the antibiotics from PMMA are typically determined through PK studies. In this context, the release kinetics of VAN were analyzed by measuring its concentration in each PMMA bead over time. The MIC of VAN for the targeted pathogen, MRSA, was considered to assess the effectiveness of the released antibiotic concentration.

Employing mathematical models for the release kinetics of antibiotics from local DDS is a valuable alternative for understanding and enhancing drug release profiles [4]. These models facilitate the interpretation and prediction of experimental outcomes, leading to efficiencies in both time and cost [4]. It is important to acknowledge that no universal model exists that fully describes all drug release mechanisms in polymeric DDS. Mathematical models provide qualitative and quantitative insights into the various factors influencing drug release, including physicochemical properties [9,10]. In this study, the Higuchi mathematical model was utilized to estimate or predict the release kinetics of antibiotics from PMMA beads.

Investigating the pharmacokinetic factors influencing a PMMA-based local DDS enables researchers to elucidate the pattern of antibiotic release, identify optimal treatment duration and concentration, and assess potential toxicity and side effects associated with the released antibiotics. This information is crucial for refining the design and efficacy of local DDS in managing DSSIs following spinal surgery.

## 2. Materials and Methods

### 2.1. Demographic Features of the Patients and Treatment Approach

This retrospective study included 23 patients who developed DSSIs following spinal surgery from June 2006 to June 2018. The patient demographic comprised 61% males and 39% females with ages ranging from 13 to 79 years and an average age of 56.1 years. The cohort underwent a range of surgical procedures including debridement for DSSIs, with treatment protocols involving the use of antibiotic-loaded PMMA beads. All surgical procedures were performed by the same neurosurgeon. 

Surgical interventions varied from multi-level fixation to single-level instrumentation and decompression. Patients were stratified into five disease categories based on the primary spinal condition diagnosed.

(i) Degenerative spinal disease was identified as the most common condition, comprising 10 cases. This was the most prevalent condition, where patients typically exhibited symptoms of spinal instability or stenosis. The surgical approach varied, with four patients undergoing multi-level spinal fusion, three patients receiving decompressive surgery (including two referral cases), and three patients treated with single-level spinal fixation. (ii) Traumatic spine injuries accounted for five cases in the cohort. Each patient in this group required multi-level fixation, which was indicative of the severe trauma-related impact, particularly thoracolumbar fractures and dislocations. (iii) The cohort also included three cases of malignant and benign spinal tumors. Two patients presented with thoracic spinal metastases requiring multi-level fixation, reflecting the complex oncological nature of these cases. Another patient with a benign lumbar neurogenic tumor underwent laminoplasty, showcasing individualized surgical planning. This denoted the oncological complexity encountered in these instances. The third case involved a lumbar benign neurogenic tumor and was performed with laminoplasty. (iv) Furthermore, our study encompassed four cases of ossification of the posterior longitudinal ligament (OPLL). Three of these cases were managed with cervical laminoplasty, and one case required multi-level fixation, indicating a spectrum of spinal canal compromise and symptomatic variation in cervical OPLL. (v) Lastly, the single case of spontaneous spinal epidural hematoma (EDH) was managed urgently with decompressive surgery and laminoplasty, emphasizing the critical and emergent nature of this condition. The diversity of conditions and the corresponding surgical interventions underscored the tailored approach to managing complex spinal pathologies and the subsequent DSSIs within this patient population. 

Patients were classified into five groups based on their primary condition: Group 1 comprised 9 patients with degenerative diseases (American Society of Anesthesiologists [ASA] Class II); Group 2 included 6 patients with traumatic spine injuries (ASA I–II); Group 3 encompassed 3 patients with spinal tumors (ASA IV); Group 4 involved 3 patients with ossification of the posterior longitudinal ligament (OPLL; ASA II–III); and Group 5 had 1 patient with spinal EDH (ASA IV). 

The identified causative pathogens were as follows: methicillin-resistant CoNS (MR-CoNS) in 10 cases, MRSA in 6 cases, unknown pathogens in 3 cases, *Staphylococcus epidermidis* in 1 case, *Enterococcus faecium-D* in 1 case, *Pseudomonas aeruginosa* in 1 case, and a mixed infection involving *E. coli* and MRSA (with *E. coli* identified as the primary pathogen). The number of PMMA beads and the chain length used were tailored to the dimensions of the infected area, ranging from 6 to 67 beads. Systemic antibiotics were prescribed tailored to organism sensitivity with durations adjusted accordingly. For resistant strains, combination or broad-spectrum antibiotics were administered. Follow-up appointments were conducted regularly over an average duration of 75 ± 52.6 months, culminating in a final assessment session focusing on clinical and radiological evaluations to monitor infection resolution and bone healing. Table 1 provides a comprehensive analysis of the clinical data for all 23 cases, including details on the pathogens identified, their antibiotic susceptibilities, and the minimum inhibitory concentrations (MICs) along with associated diseases.

### 2.2. PMMA DDS Beads Formulation

In all cases, PMMA was employed, meticulously crafted by hand, and under aseptic conditions within a sterile operating room during the surgery. Typically, two 1 g vials of VAN powder were mixed for most cases (19 cases) and combined with an additional 2 g of ceftriaxone (CRO) in 3 cases. For *P. aeruginosa* infection, 2 g of ceftazidime (CAZ) was used, while 2 g of ertapenem (ETP) was employed for *E. coli* infection. These antibiotics were thoroughly mixed with 40 g of PMMA (DePuy CMW3 Gentamicin, Raynham, MA, USA) in sterile dishes. The solvent volume was 10 mL for a 20 g PMMA dosage and 20 mL for a 40 g PMMA dosage. When the mixture reached a non-adhesive, malleable consistency, beads approximately 1 to 1.5 cm in diameter were manually formed, averaging about 10 beads per sterile stainless-steel wire. The number of wires used was contingent on the size of the surgical wound, varying from 2 to 3 wires to as many as 6 wires. Each bead contained 48–72 mg of VAN as the active ingredient. The weight of each bead, when loaded with VAN and PMMA, ranged from 616 to 2000 mg, calculated using the formula Weight = Volume × Density, considering the combined volume and density of VAN and PMMA.

### 2.3. Beads Stringing and Antibiotic Concentrations

The prepared PMMA beads were strung onto a stainless-steel wire with the ends securely twisted to form a firm knot. We employed two distinct sets of antibiotic concentrations: 48 mg each of VAN, CRO, and CAZ, which were incorporated into PMMA beads with a diameter of 1 cm. In contrast, the second set consisted of 1.5 cm diameter beads, containing 72 mg of VAN only. Subsequently, drug release kinetics were analyzed about the concentration and size of the beads for the different types of antibiotics used. 

### 2.4. Application of Mathematical Model for the Estimation of Release Kinetics from Antibiotics-Loaded PMMA Beads

#### 2.4.1. Surface Area Considerations

In this study, the surface area of the PMMA beads, an important determinant in drug release and absorption, was precisely calculated. Beads with a diameter of 1 cm had a surface area of 3.14 cm^2^, while those with a 1.5 cm diameter exhibited a larger surface area of 7.06 cm^2^. The pharmacokinetic properties of the antibiotics were then analyzed about these variations in bead surface area.

#### 2.4.2. Mathematical Modeling in Drug Delivery Systems

Mathematical modeling was applied to our local DDS to elucidate the complex interactions between variables affecting drug release, thereby evaluating the safety and effectiveness of localized drug therapy. These mathematical models are known to provide insights for predicting and controlling drug release, which is crucial for treating bone infections or DSSIs. They also aid in analyzing factors affecting drug release, including the materials used, drug concentrations, and mixing methods. The application of these models enables the customization of drug release profiles to meet specific treatment needs, facilitating more accurate prediction of treatment outcomes.

#### 2.4.3. Employment of the Higuchi Model

Several mathematical models are available for analyzing drug release kinetics. In this study, the Higuchi model, represented by the formula Q = [Dε/τ (2C − εCs) Cst]^1/2^, was used to calculate drug release kinetics from PMMA beads [11]. In this equation, Q represents the amount of drug released per unit area over time t; D is the diffusion coefficient of the drug in the liquid within the porous structure; ε denotes the porosity of the porous framework; τ is the tortuosity factor of the capillary system; Cs is the solubility of the drug in the matrix substance; and C is the initial drug concentration in the matrix. Utilizing data from published literature employing the same matrix for delivering antibiotics to treat SSIs, we input values into the equation using Wolfram Mathematica with time (t) measured in days. The diffusion coefficients of VAN, CRO, and CAZ in water at 37 °C, the porosity of the PMMA matrix, the tortuosity of the matrix, and the solubility of the antibiotics in PBS were among the parameters considered.

For this study, data from the published literature using the same matrix for treating SSI were utilized for the parameter values required for the application of the release dynamic mathematical model. According to that, the diffusion coefficient of VAN in an intervertebral disc is (D) = 7.94 ± 2.00 × 10^−12^ m^2^/s [12], the diffusion coefficient of VAN in water at 37 °C was 2.83 × 10^−10^ m^2^/s [13], and the diffusion coefficient of CRO and CAZ in water at 37 °C was 2.77 × 10^−12^ m^2^/s 2.33 × 10^−12^ m^2^/s, respectively. The porosity (ε) of the PMMA matrix was 0.5 ± 0.2% [14], with a pore size of 300–500 μm [15], and the tortuosity (τ) was set at 1.24 [16]. In phosphate-buffered saline (PBS), the solubility of VAN, CRO, and CAZ were 5 mg/mL, 10 mg/mL, and 10 mg/mL, respectively. These parameter values were integrated into the equation using Wolfram Mathematica with time (t) expressed in days.

#### 2.4.4. Bead Volume Calculation

The volume (V) of each bead was calculated using V = (πd^3^)/6 or V = (4/3) πr^3^, where π is about 3.14159, r is the sphere’s radius, and d is the diameter. The densities used were PMMA at 1.18 g/cm^3^, VAN at 1.42 g/cm^3^, and water at 1 g/cm^3^. 

Based on our calculations, the volume of each bead is 0.5236 cc. In our study, we used 20–40 g of PMMA, 2 g of VAN, and 10–20 mL of water. With densities of PMMA (1.18 g/cm^3^) and VAN (1.42 g/cm^3^), the total composite volume ranged from 22 to 42 cm^3^, and each bead contained 48–72 mg of antibiotics. Three groups were formed to analyze the effect of PMMA bead quantity on release kinetics: Group A (6–17 beads), Group B (20–30 beads), and Group C (40–67 beads), enabling a dose–response trend analysis based on bead quantity. 

Graph Pad Prism version 10.1.2 for Windows software (GraphPad Software, San Diego, CA, USA) was employed to analyze the statistical data. The figures were generated for graphic illustration and visualization.

## 3. Results

### 3.1. Clinical Outcome

In this study, the patient cohort, with an average age of 56.1 years (median 58.5) ranging from 13 to 79 years, presented a male-to-female ratio of 14:9. The lumbar vertebra was identified as the most common site of infection in 12 patients (52%), which was followed by the cervical vertebra in six cases (26%) and the thoracic vertebra in five cases (22%). The average interval between the initial surgery and the first implantation of the DDS was 28.7 days (median 18 days). Degenerative spine disease showed the highest SSI incidence at 43.5%, which was followed by traumatic spine injuries at 21.7%. Spinal tumors and OPLL contributed 13% and 17.3% of SSIs, respectively, while spontaneous epidural hematoma (EDH) accounted for 4.3%. Lumbar surgery had the highest SSI incidence (52%), which was followed by thoracic (26%) and cervical surgery (22%). Multi-level fixation had a higher SSI incidence (78%) compared to single fixation (22%). The results illustrated the type of surgery and severity of diseases and relevant factors affecting SSI incidence. Over a follow-up period of 75 ± 52.6 months, all cases were successfully treated with no recurrences or complications. Notably, no signs of kidney or liver dysfunction were observed. Table 1 categorizes patients based on the associated spinal diseases, providing a clear linkage between pathogen type and clinical diagnosis. In addition, it provided an extensive overview of the pharmacokinetics involved, featuring both daily and cumulative drug release profiles derived from mathematical modeling.

### 3.2. Microbiological Outcome

This study identified various causative pathogens including MRSA sensitive to VAN in 19 cases, 1 case of *P. aeruginosa* sensitive to CAZ, and 3 cases with unknown pathogens sensitive to VAN and CRO. The most frequently occurring pathogen was MR-CoNS in 10 cases (43.5%), which was followed by MRSA in 6 cases (26%). Additionally, three cases (13%) were attributed to unknown pathogens. One case each (4.35%) was linked to *S. epidermidis*, *E. faecium-D*, *P. aeruginosa*, and *E. coli* (mixed infection). The MIC values determined for the antibiotics against various pathogens included MR-CoNS (≥8 µg/mL), MRSA (≤1 µg/mL), *P. aeruginosa* (≤1 µg/mL), *E. faecium-D* (≤0.5 µg/mL), *S. epidermidis* (≤0.5 µg/mL), and MRSA (≤0.5 µg/mL). These findings highlighted pathogen sensitivity to antibiotics and supported the efficacy and safety of the localized treatment strategies implemented in this study.

### 3.3. Pharmacokinetic Interpretation

#### 3.3.1. Mathematical Modeling Interpretation

The Higuchi model was employed to analyze antibiotic release from PMMA beads over 42 days in 23 cases, as detailed in Table 2. The average cumulative release values were as follows: VAN at 3021.45 µg/cm^2^, CRO at 4204.14 µg/cm^2^, and CAZ at 2741.85 µg/cm^2^ for 1 cm diameter beads and VAN for 1.5 cm diameter beads at 3700.83 µg/cm^2^. The corresponding median values were VAN at 3189 µg/cm^2^, CRO at 4437.5 µg/cm^2^, CAZ at 2898.5 µg/cm^2^ for 1 cm diameter beads, and VAN at 3906.5 µg/cm^2^ for 1.5 cm diameter beads. The mean daily release rates were VAN at 81.1 µg/cm^2^, CRO at 104.9 µg/cm^2^, and CAZ at 69.8 µg/cm^2^, with median daily values of 75 µg/cm^2^ for VAN, 104.5 µg/cm^2^ for CRO, and 66.5 µg/cm^2^ for CAZ. These findings provided insights into the variable rates of antibiotic release from PMMA beads.

#### 3.3.2. Release Kinetics

Table 2 presents a detailed analysis of the release kinetics for various antibiotics, utilizing a mathematical model. The table outlines key metrics for each antibiotic, including mean and median cumulative release (CR) in µg/cm^2^, mean and median daily release rate (DRR) in µg/cm^2^, initial or burst concentration in µg/cm^2^ with its percentage, lag phase concentration in percentage, plateau concentration in percentage and µg/cm^2^, and the daily release percentage up to 42 days. The release kinetics from the PMMA beads showed a distinct pattern over the study period. Initially, there was a burst phase observed within the first 1–2 days with peak concentrations of VAN reaching 687 µg/cm^2^ (15.43%), CRO reaching 957 µg/cm^2^ (15.42%), and CAZ reaching 624 µg/cm^2^ (15.41%). This was followed by a lag phase starting between days 2 and 3, where there was a decrease in the release rates to 4.91% for VAN, 4.92% for CRO, and 4.91% for CAZ. Subsequently, a plateau phase was evident from days 5 to 7, with the concentration of CRO stabilizing at 3.17% and the concentration of CAZ and VAN ranging between 3.15% and 3.16% and 3.36% for VAN 1.5. The release then proceeded with a consistent or steady daily release rate of 1.81% for VAN, 1.69% for CRO, 1.73% for CAZ and 2.39% for VAN 1.5 resulting in average concentrations of 81 µg/cm^2^ for VAN, 105 µg/cm^2^ for CRO, 70 µg/cm^2^ for CAZ and 131 µg/cm^2^ for VAN 1.5 maintained up to day 42. This release pattern provided valuable insights into the dynamic nature of drug release kinetics, offering a deeper understanding of antibiotic behavior over extended periods.

#### 3.3.3. Parameters Affecting Release Kinetics

This prospective cohort study investigates various physicochemical parameters influencing release kinetics, focusing on the characteristics of beads, diffusion coefficients of specific drugs (VAN, CRO, and CAZ), and matrix properties. Beads ranging in diameter from 1 to 1.5 cm, forming chains of 6 to 40 beads, were studied, resulting in lengths between 17 and 93 cm. Each bead exhibited a surface area of approximately 3.14 to 7.06 cm^2^.

The diffusion coefficients (diffusivity) of VAN, CRO, and CAZ, crucial for drug release, were assumed from the published literature as follows: VAN at 2.83 ± 2.00 × 10^−6^ cm^2^/s, CRO at 2.74 ± 2.00 × 10^−6^ cm^2^/s, and CAZ at 2.83 ± 2.00 × 10^−6^ cm^2^/s [13,17]. Matrix properties, specifically porosity and tortuosity, were found to impact release kinetics. The matrix in this study was assumed to have a porosity of about 0.5 ± 0.2% [14] and a tortuosity of 1.24 [16]. The concentration of the drug in the matrix was set at 5%, significantly influencing release kinetics. Solubilities of VAN, CRO, and CAZ in phosphate-buffered saline (PBS) were 5 mg/mL, 10 mg/mL, and 5 mg/mL, respectively. Drug release measurements were conducted over 42 days, from day 1 to day 42 [8,18].

Bead size and polymerization time were identified as additional factors affecting drug release from PMMA beads [19]. Beads ranging from 1 to 1.5 cm in diameter were examined, revealing that larger beads exhibited increased drug release, primarily due to their larger total surface area.

Utilizing a mathematical equation, the estimated elution on day 42 for PMMA beads was categorized into three groups based on release amount in mg/cm^2^. Category 1 showed a mean/median concentration of 76.97 mg/cm^2^/89.16 mg/cm^2^ with specific cases exhibiting lower values (cases 4, 7, 13, 16 to 20, and 22). Category 2 exhibited a mean/median concentration of 162.11 mg/cm^2^/178.32 mg/cm^2^ with moderately higher values seen in cases 1, 3, 6, 8, 9, 11, 14, and 23. Category 3 displayed a mean/median concentration of 260.32 mg/cm^2^/249.64 mg/cm^2^ with the highest values in cases 2, 5, 10, 12, 15, and 21.

#### 3.3.4. Cumulative Release Kinetics

The cumulative drug release over 42 days provided valuable insights for a single bead with a 1 cm diameter, yielding the following results: 4458 µg/cm^2^ (9.3%) for VAN, 6202 µg/cm^2^ (12.9%) for CRO, 4045 µg/cm^2^ (8.4%) for CAZ, and 5460 µg/cm^2^ (7.6%) for VAN in a single bead with a 1.5 cm diameter. The antibiotic release kinetics, demonstrated in Figure 1, showed distinctions between VAN, CRO, and CAZ over the 42 days: CRO vs. VAN (32%), CRO vs. CAZ (42%), and VAN vs. CAZ (9.7%). Expanding the bead size from 1 to 1.5 cm for VAN increased the surface area by 20%: from 3.14 to 7.06 cm^2^. This modification led to a 5.46 mg (22.45%) increase in total cumulative release but resulted in a slight reduction in the fraction released from the total concentration by 1.7%. These results revealed the complex relationship between antibiotics, bead size, and release dynamics, offering key insights for optimizing DDS. Statistical significance was confirmed for the differences in cumulative release among the antibiotics (*p* < 0.001, as shown in Figure 1, using a one-way ANOVA test). The area under the curve (AUC) values were as follows: 172,995 μg·day/cm^2^ for CRO, 124,329 μg·day/cm^2^ for VAN, 152,284 μg·day/cm^2^ for VAN in the 1.5 cm diameter bead, and 112,824 μg·day/cm^2^ for CAZ.

#### 3.3.5. Daily Release Kinetics

The daily release of antibiotics from a single PMMA bead exhibited distinct phases during a one-week interval and over a total of 42 days in Figure 2a,b. Initially, a burst phase was observed within the first 1–2 days, which was characterized by peak concentrations of 687 µg/cm^2^ for VAN, 957 µg/cm^2^ for CRO, and 624 µg/cm^2^ for CAZ. Approximately 1.99% of the total CRO, 1.3% and 1.4% of CAZ, and VAN were released during this phase (days 2–3: VAN: 1.05%, CRO: 1.4%, and CAZ: 0.94%). Following this burst phase, a lag phase occurred between days 2 and 3, during which the release rates declined to 4.91% for VAN, 4.92% for CRO, and 4.91% for CAZ. Subsequently, a plateau phase emerged from days 5 to 7, with stable release rates: CRO at 3.17%, CAZ/VAN between 3.15% and 3.16%, and VAN 1.5 at 3.36%. This phase was followed by a consistent daily release contributing 1.8% for VAN, 1.69% for CRO, 1.73% for CAZ, and 2.39% for VAN 1.5 to the overall antibiotic concentration, resulting in average concentrations of 81 µg/cm^2^ for VAN, 105 µg/cm^2^ for CRO, 70 µg/cm^2^ for CAZ and 131 µg/cm^2^ for VAN 1.5 extending up to day 42. These findings provided valuable insights into the long-term dynamics of antibiotic release.

#### 3.3.6. Comparison of Cumulative Antibiotic Release Based on the Number of Beads

The impact of bead quantity on the cumulative antibiotic release was systematically represented in Figure 3a–c, which delineated the antibiotic release patterns across three distinct groups: A, B, and C. Each of these groups was characterized by different quantities of beads. 

Group A, comprising 6 to 17 beads with diameters of both 1.0 cm and 1.5 cm, showed a release range of 26.7 to 86.8 mg/cm^2^. Notably, 14 beads of CRO released a greater amount of antibiotics compared to 17 beads of VAN, highlighting the complexity of release kinetics that extend beyond basic determinants of concentration factors. Additionally, when comparing bead diameters—1.0 cm versus 1.5 cm for VAN (6 beads each)—the larger beads exhibited a higher cumulative release, indicating complexities in release kinetics that surpass the simplistic factors of concentration and bead diameter (Figure 3a). 

Group B, consisting of 20 to 30 beads of 1.0 cm diameter each, exhibited a release range between 89.1 and 186 mg/cm^2^. The distinct release profiles of CRO, VAN, and CAZ with 30 beads were evaluated, considering their solubility and diffusivity, which were crucial for achieving the desired therapeutic effects. The overall cumulative release over 42 days, considering a total of 30 beads with a 1 cm diameter each, is as follows in Group B: 133.7 mg/cm^2^ for VAN, 186 mg/cm^2^ for CRO, and 121 mg/cm^2^ for CAZ. These values highlighted significant differences in cumulative release among the antibiotics, with CRO exhibiting a 39.25% higher release compared to VAN, a 53.72% higher release compared to CAZ, and VAN demonstrating a 10.5% higher release compared to CAZ (*p* < 0.001, as shown in Figure 3b, using a one-way ANOVA test). These differences demonstrated the varying release profiles and behavior of these antibiotics, providing essential insights for DDS optimization in Group B. 

Finally, Group C, with 40 to 67 beads, demonstrated a release range from 178.3 to 298.6 mg/cm^2^ (Figure 3c). This comprehensive analysis effectively demonstrated the significant impact of bead quantity on antibiotic release, elucidating the intricate dynamics of antibiotic delivery from PMMA beads and underscoring their critical implications for enhancing therapeutic efficacy. 

The findings were visually represented in Figure 3, which illustrates the influence of bead quantity on cumulative antibiotic release and provides a comparative analysis of the release profiles for each antibiotic: CRO, VAN, and CAZ. Notably, the reduced release observed for VAN and CAZ antibiotics could be attributed to their lower solubility and diffusivity within the PMMA beads compared to CRO. To achieve the desired therapeutic effect, one augmentation option considered was to increase the doses of antibiotics with lower solubility and diffusivity while also optimizing the bead diameter. This approach had the potential to enhance antibiotic release kinetics and enable the customization of drug delivery to meet specific therapeutic requirements.

## 4. Discussion

Mathematical models have proven to be invaluable tools in understanding and predicting various aspects of DDS. They play a crucial role in developing pharmaceutical formulations, assessing drug release mechanisms in both laboratory and real-world settings, and essentially in formulating the most effective strategies for innovative systems [20]. Creating mathematical models to understand how drugs are released over time is crucial for studying the processes that control this release [21]. Many comprehensive reviews have been written about mathematical modeling in different drug delivery systems, such as biodegradable polymeric systems [22], dissolution-controlled drug delivery systems [23], microsphere delivery systems [24], and hydrogel networks [25]. Generally, the main ways drugs move out of polymeric matrices are through diffusion, erosion, and degradation [26]. This paper explores models for materials with more complex compositions and structures.

However, it is crucial to recognize their limitations, especially in scenarios involving complex dynamics. One significant limitation is their inability to accurately estimate fluctuations in drug release and peak concentrations (C_max_) from delivery systems. This shortcoming can pose challenges in achieving consistent therapeutic outcomes, as variations in drug release may lead to reduced effectiveness or unforeseen side effects.

Moreover, mathematical models encounter difficulties in predicting elution kinetics for combined multi-regimen antibiotic therapies. The intricate interplay between multiple antibiotics and their respective release profiles within a single delivery system complicates accurate prediction. Additionally, the prediction of drug release can be affected by challenges in estimating fluid accumulation at the wound site. This discrepancy between the actual and predicted fluid levels can significantly alter the local drug concentration, subsequently influencing the release kinetics. These challenges underscore the importance of accurate drug dosing for successful treatment outcomes, particularly in clinical scenarios where precise medication management is crucial. 

To enhance the reliability of the mathematical models in DDS, integrating in vitro study data is crucial. These data not only enhance the accuracy of modeling but also provide a means to validate the model’s predictions against real-world scenarios.

The effective management of DSSIs requires the identification of causative pathogens and their antibiotic resistance profiles. Developing potent and effective multi-drug regimens, especially for highly resistant pathogens, poses significant challenges. The integration of both systemic and local antibiotic administration has emerged as a promising strategy, showing improved results in reducing the side effects associated with DSSIs. Additionally, there is an increasing shift toward adaptive local DDS, which aligns with the principles of personalized medicine. This approach focuses on tailoring treatment to the individual needs of patients, considering specific bacterial resistance patterns and patient characteristics. This tailored approach aims for efficient drug delivery while minimizing the side effects commonly associated with DSSIs.

### 4.1. Antibiotic Release from PMMA Beads

The integration of antibiotics into PMMA employs both manual and vacuum mixing methods. Manual mixing, typically used in orthopedics, tends to result in higher antibiotic elution, which is primarily due to increased porosity in PMMA. Studies indicate that hand-mixed cement releases significantly more antibiotics than commercially mixed cement. Specifically, VAN elution was five-fold higher in hand-mixed cement, and gentamicin elution was two-fold greater [27]. However, the choice of mixing method has implications for antibiotic release. Manual mixing results in higher antibiotic concentrations, while vacuum mixing reduces the porosity and burst release of antibiotics, contributing to improved cement quality. Nevertheless, vacuum mixing may raise concerns about thermal conductivity and tissue necrosis. There are conflicting reports regarding the relationship between antibiotic release, antibiotic loading, and mixing methods [28]. These results show how the mixing method affects antibiotic release, impacting infection treatment effectiveness and patient safety.

Limited data exist on the C_max_ of antibiotics released from PMMA beads. Previous research has indicated peak concentrations of 198 µg/mL on day one, reducing to 23 µg/mL after 13 days [29]. An alternative formulation of PMMA with 2 g VAN and 22 g xylitol achieved peak concentrations exceeding 240 µg/mL on day one, maintaining over 12.5 µg/mL by day 10, surpassing the MIC for sensitive strains by 120 times and 6.25 times, respectively [30]. In our study, a C_max_ of 687 µg/cm^2^ was estimated using a mathematical model. When evaluating antibiotic effectiveness in PMMA-based local DDS, it is crucial to consider factors such as therapeutic levels, duration of efficacy, local tissue toxicity, and pathogen MIC. Antibiotic effectiveness is determined by the duration over which concentrations remain above the MIC with a higher area under the curve (AUC)/MIC ratio indicating enhanced efficacy. A specific AUC/MIC ratio of 400 has been linked with successful MRSA eradication in adult lower respiratory tract infections [28]. Treatment duration varies but averages 42 days for VAN when administered via PMMA spacer and beads [8,18]. The critical parameters for evaluating antibiotic efficacy, especially in local DDS like PMMA beads, are MIC and C_max_, recognizing the variability in MIC among different bacterial strains. AE, as determined by the AUC/MIC ratio, plays a pivotal role in optimizing treatment duration and ensuring effectiveness in complex infections.

### 4.2. Impact of Antibiotic Levels in Antibiotic-Loaded Bone Cement (ALBC)

Elevated concentrations of antibiotics in ALBC can have adverse effects on tissue cells and bone formation. Gentamicin exhibits toxicity at concentrations as low as 200 µg/mL. In contrast, vancomycin (VAN) and tobramycin are less toxic, but their toxicity becomes significant at concentrations exceeding 2000 µg/mL. VAN can induce cell death at 5000 µg/mL, and tobramycin impacts cell replication beyond 500 µg/mL, leading to significant cell death at concentrations of 5000 µg/mL. Prolonged exposure and intracellular accumulation can lower these toxicity thresholds, as demonstrated in cases of acute renal failure due to the systemic absorption of high-dose ALBC spacers containing tobramycin and VAN [28]. This highlights the need for a careful calibration of antibiotic levels in ALBC to ensure efficacy without compromising tissue health and bone integrity.

The effect of antibiotics on cellular health is a critical research area, especially given the multifaceted impact of high concentrations on cell survival, proliferation, metabolism, and differentiation. Certain antibiotics, such as Rifampin and minocycline, have been shown to diminish osteoblast numbers and activity, whereas others like amikacin and vancomycin (VAN) exhibit lower cytotoxicity, particularly at higher concentrations [31,32]. In orthopedic procedures aimed at preventing and treating bone infections, two dosages of antibiotic-impregnated cement spacers are typically used: low dose and high dose. The high-dose cement is often preferred for established infections [4]. Studies have demonstrated that patients receiving 10.5 g VAN and 12.5 g gentamicin in spacers, alongside intravenous antibiotics, did not experience systemic toxicity [33]. For conditions such as open fractures and osteomyelitis, recommended dosages include 4 g VAN and 3.6 g tobramycin per 40 g PMMA cement [34]. However, caution is advised when using high-dose ALBC spacers containing multiple antibiotics, as cases of acute renal failure have been reported [35]. High-loading doses of antibiotics in PMMA can also affect the mechanical strength of bone cement, with a recommended limit of 5%, not exceeding 10–15%. Several FDA-approved ALBCs are available, including Cobalt g-HV, Palacios G, DePuy 1, Cemex Genta, VersaBond AB, Simplex P, Biomet Refobacin Cement R, and Palacios R+G p [28]. This knowledge empowers healthcare professionals to make informed decisions, ensuring safe, effective antibiotic use in infection treatment, and preserving bone health in orthopedic procedures.

### 4.3. Combination Therapy in Chronic Osteomyelitis

Combination therapy has been emerging as a more effective approach than monotherapy in the treatment of chronic osteomyelitis [36]. Research by Evans and Nelson, for instance, used a rabbit model to explore optimal treatment strategies. They discovered that a comprehensive approach combining systemic antibiotics with local delivery, such as debridement, gentamicin beads, and intravenous CRO, achieved a 100% infection control rate. In contrast, monotherapies like gentamicin beads alone or intravenous CRO alone yielded lower control rates of 79% and 92%, respectively [37]. Despite the promising outcomes of combination therapy, a significant challenge in managing DSSIs is the lack of consensus regarding the ideal duration of antibiotic treatment, leading to uncertainty in clinical decision-making and potentially affecting treatment outcomes. Further research and consensus-building are needed to establish clear guidelines for the duration of antibiotic therapy in DSSIs cases [35]. Recent guidelines endorse VAN as the preferred antibiotic for treating DSSIs, underscoring its effectiveness against such infections [4]. This choice highlights VAN’s effectiveness in addressing DSSIs, which is a critical factor for healthcare providers when planning treatments. With these insights into combination therapy benefits and ongoing debates about antibiotic duration, along with the favored antibiotic for DSSIs, healthcare professionals can now make more informed decisions in managing chronic osteomyelitis and related infections, potentially improving patient outcomes. 

The current use of antibiotics-loaded PMMA, combining agents like gentamicin, tobramycin, and VAN, aims to prevent or treat bacterial infections due to their broad-spectrum efficacy and bacteria-fighting mechanisms. However, the use of gentamicin in bone cement has led to the emergence of gentamicin-resistant *Staphylococcus aureus* strains in medical practice [38]. Thus, the incorporation of antibiotics into PMMA can contribute to antibiotic resistance, particularly with long-term exposure to ALBC, posing an emerging threat in medicine [28]. The selection of antibiotics for ALBC should consider their broad-spectrum coverage, mechanisms of action against bacteria, and the risk of resistance development. Key considerations in developing ALBC include mechanical properties, antibiotic elution, and bone ingrowth [28].

Overall, mathematical models play an important role in understanding DDS dynamics but have limitations in predicting fluctuations and managing complex personalized treatment scenarios. Empirical data and clinical observations are essential to enhance model accuracy. The choice of mixing methods in ALBC impacts release patterns and patient safety, and an understanding of antibiotic toxicity and dosages is crucial for maintaining bone health in infections. Combination therapy, particularly systemic and local antibiotic delivery, shows promise for chronic osteomyelitis, but the ideal duration of antibiotic treatment remains debatable. Furthermore, while ALBC is effective against infections, it poses the risk of contributing to antibiotic resistance. Healthcare professionals must balance these benefits and risks when using ALBC materials. This comprehensive understanding empowers healthcare professionals to make informed decisions, ensuring the use of effective and safe antibiotics in an appropriate manner.

The limitation of this study: This study acknowledges several limitations. The absence of in vitro data, which may differ from real-world scenarios, could impact the robustness of the findings. Additionally, the small sample size may limit the generalizability of the results. The lack of validation and verification raises questions about the accuracy and reliability of the conclusions. These limitations highlight the need for cautious interpretation of the study’s results and emphasize the potential for further research to address these issues. 

## 5. Conclusions

In conclusion, the use of antibiotics-loaded PMMA loaded with antibiotics for localized drug delivery presents a promising strategy in the treatment of DSSIs. This approach not only effectively reduces the typical side effects associated with systemic antibiotic administration but also maintains optimal drug concentrations for extended periods, which is particularly beneficial in cases with compromised glomerular filtration rate (GFR).

While mathematical modeling is instrumental in predicting the release kinetics of single antibiotics from carriers, it is essential to recognize its limitations in accurately predicting the elution kinetics of multi-regimen antibiotics. Advancements in this area necessitate robust validation employing authentic release kinetics methods, taking into account pharmacokinetic parameters and considering biological and environmental factors. Additionally, investigating nonlinear release behavior through both in vivo and in vitro studies is crucial.

Future research should be directed toward optimizing drug release profiles. Key focus areas include controlling the initial burst of medication, achieving sustained release, ensuring safe tissue concentrations, and developing biodegradable carriers. Such advancements are expected to enhance the efficacy of localized drug delivery systems (DDSs), thereby improving the management of infections and patient outcomes.

## Figures and Tables

**Figure 1 pharmaceutics-16-00094-f001:**
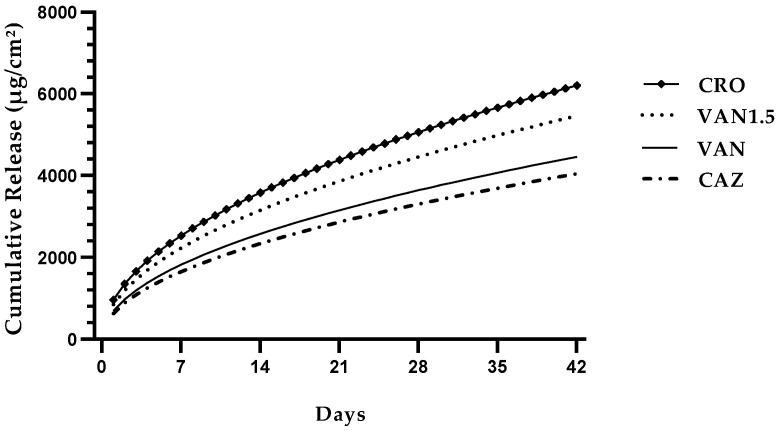
Cumulative antibiotic release from a single PMMA bead. The differences in cumulative release between CRO vs. CAZ (53.3%), CRO vs. VAN (39.1%), and 1.5 cm diameter VAN over 1.0 cm diameter VAN (22.1%) were all statistically significant (*p* < 0.001, as determined by the one-way ANOVA test).

**Figure 2 pharmaceutics-16-00094-f002:**
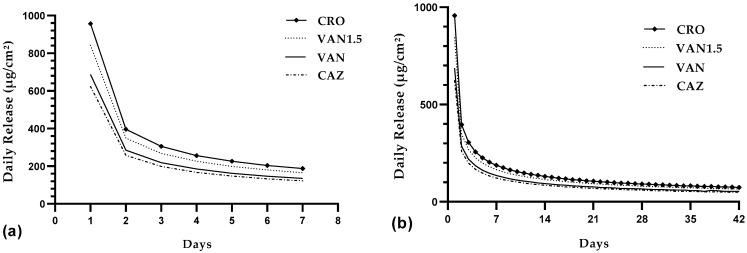
Daily antibiotic release from a single PMMA bead. Three distinct phases of daily release were observed during the first week (**a**), which are then followed by a consistent daily release pattern extending from day 8 to day 42 (**b**). The mean daily release rates over 42 days are 1.81% for VAN, 1.69% for CRO, 1.73% for CAZ, and 2.39% for VAN 1.5.

**Figure 3 pharmaceutics-16-00094-f003:**
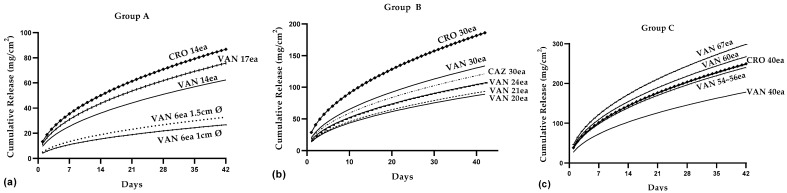
Impact of bead quantity on cumulative antibiotic release.

**Table 1 pharmaceutics-16-00094-t001:** Comprehensive clinical summary including daily and cumulative release kinetics.

Pathogen	Abx	Ds Entities	Pts	ASA Score	Implants	Single vs. Multiple	MIC (µg/mL)	Daily Release Min to Max (µg/cm^2^)	Cumulative Release Min to Max (µg/cm^2^) in Group B (30ea)	Case No.
MRCoNS	VAN	4 Deg. Ds, 2 traumas, 2 OPLL, 2 Mets	10	I~IV	Yes	9 Multiple 1 Single	≥8	54~687 (106.1 ± 16.0)	20,610~133,700 (90,640 ± 4783)	#1, 7~9, 12, 15, 16, 18, 19
MRSA	VAN	4 Trauma 2 Deg. Ds	6	I~III	Yes	4 Multiple 2 Single	≤1	54~687 (106.1 ± 16.0)	20,610~133,700 (90,640 ± 4783)	#2, 3, 5, 6, 13, 14
Unknown -> *P. aeruginosa*	VAN→ CAZ	Deg. Ds (1 referrals)	1	II	No	Multiple	≤1	49~624 (96.3 ± 14.5)	18,720~121,400 (82,260 ± 102,600)	#17 (30ea) *
*E. faecium-D*	VAN	Spine Tumor	1	IV	No	Multiple	≤0.5	54~687 (106.1 ± 16.0)	14,430~93,620 * (63,450 ± 3348)	#20 (21ea) *
*S. epidermidis*	VAN	Spont EDH	1	IV	Yes	Multiple	≤0.5	54~687 (106.1 ± 16.0)	5520~32,760 * (22,210 ± 1172)	#22 (6ea) *
*E. coli* & MRSA	ETP—> VAN	Deg. Ds (1 referral)	1	IV	Yes	Multiple	≤0.5	54~687 (106.1 ± 16.0)	27,480~178,300 * (120,900 ± 6377)	#23 (40ea) *
Unknown	VAN + CRO	2 Deg. Ds, and 1 OPLL	3	I~II	Yes	Multiple	-	74~957 (147.7 ± 22.3)	28,710~186,100 (82,260 ± 28,130)	#4, 10, 11

Note: Abx, antibiotics; Ds, disease; Pts, number of patients; Implants, surgical instrumentation; MIC, minimum inhibitory concentration; Case No., #, individual case number; () *, number of beads in the individual case; Deg. Ds, degenerative spine disease; OPLL, ossification of posterior longitudinal ligaments; Mets, metastases; EDH, epidural hemorrhage; referral, a patient who was referred from another hospital; VAN, vancomycin; CRO, ceftriaxone; CAZ, ceftazidime; ETP, ertapenem.

**Table 2 pharmaceutics-16-00094-t002:** Analysis of release kinetics using a mathematical model.

Antibiotic Laden Bead	Burst Release (1~2 Days)	Lag Phase (3~4 Days)	Plateau Phase (5~7 Days)	Mean DRR (~42 Days)	Mean CR for 42 Days	*p*-Value (ANOVA)
VAN	687 (15.43%)	202 (4.91%)	148 (3.32%)	81 (1.81%)	3021.45	
CRO	957 (15.42%)	280.5 (4.92%)	198 (3.19%)	105 (1.69%)	4204.14	
CAZ	624 (15.41%)	183 (4.91%)	134 (3.31%)	70 (1.73%)	2741.85	*p* < 0.001
VAN (1.5 cm)	842 (15.42%)	247 (4.90%)	184 (3.36%)	131 (2.39%)	3700.83	

Note: VAN, vancomycin; CRO, ceftriaxone; CAZ, ceftazidime; VAN, CRO, and CAN in PMMA beads with a diameter of 1.0 cm each; VAN (1.5) represents vancomycin in beads with a diameter of 1.5 cm; DRR, daily release rate (µg/cm^2^); burst release, lag phase and plateau phase, mean DRR during the phases (µg/cm^2^); CR, cumulative release over 42 days (µg/cm^2^).

## Data Availability

Data are contained within the article.

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
