# Peer review of "Pharmacokinetic Interpretation of Applying Local Drug Delivery System for the Treatment of Deep Surgical Site Infection in the Spine"

_pharmaceutics, 2024, doi:10.3390/pharmaceutics16010094_

Round 1
Reviewer 1 Report
Comments and Suggestions for Authors
Dear authors,
Your paper is very nice written article and concerns important hospital postoperative infections.
Many clinical microbiologists do not support inappropriate and no useful use of antimicrobial drugs, which, in addition to having no therapeutic effect, also affects the unwanted occurrence of bacterial resistance to antibiotics. This approach enables, on the one hand, the treatment of infection and, on the other hand, the “saving” an effective antibiotics.
I recommend it for publication with minor revision.
General comment
The most important is in my opinion and regarding the paper:
In part where you highlighted limitation of paper regarding the number of patients (23), you have to add the consideration of the number of operated persons and the variety of spine operations. Besides, you present patients with operations due to degenerative diseases of the spinal column (9 in total) - at the same time, you did not say what types of degenerative operations (disc hernia, stenosis of the spinal canal, spinal scoliosis); also spinal column trauma (on one or more levels, what types of traumatic reconstruction?; Spinal tumors - also what type (malignant, benign, etc.) and origin (bone, joint, connective, radicular)? ; Ossification of the posterior longitudinal ligaments at how many levels?; Spinal epidural hematoma (traumatic or spontaneous); It also remains unclear whether the infection was localized to the bony structures of the spinal vertebrae or whether it affected other elements as well (connective tissue, ligaments, spinal roots, deep muscles?; Wound debridement - it should be described in detail - did it affect only the skin and subcutaneous tissue or deeper elements (for example, muscle tissue and fascia).
-you did not mention anywhere in the paper the approval of the institution's Ethics committee for research
Specific comment:
-In all text of article species of bacteriae were not written italic.
-You have to check sentences in many parts they beginning with small letter and confusing reader
-Also you have to check all abbreviations to be explained when you firstly mentioned them
Comments on the Quality of English LanguageMinor editing of English language required
Author Response
Dear Editor and Reviewer,
Thank you for giving me the opportunity to submit a revised draft of my manuscript. I appreciate the time and effort that editor and the reviewers have dedicated to providing your valuable feedback on my manuscript. I am grateful to the reviewers for their insightful comments on my paper. We have been able to incorporate changes to reflect most of the suggestions provided by the reviewers. We have highlighted the changes within the manuscript. Here is a point-by-point response to the reviewers’ comments and concerns.
Response to Reviewer Comments
- Response to Reviewer 1
II-1. General comment-The most important is in my opinion and regarding the paper:
- In the part where you highlighted the limitation of the paper regarding the number of patients (23), you have to add the consideration of the number of operated persons and the variety of spine operations.
- Response: Thank you for your insightful feedback. Our study is a retrospective clinical analysis focused on the surgical treatment of deep surgical site infections following spine surgery, conducted by a single neurosurgeon. We employed pharmacokinetic mathematical modeling and antibiotic release evaluations in a cohort series. The study encompasses a range of spinal surgeries, with detailed case descriptions provided in Table 1 and further elaborated upon in the text.
- Besides, you present patients with operations due to degenerative diseases of the spinal column (9 in total) - at the same time, you did not say what types of degenerative operations (disc hernia, stenosis of the spinal canal, spinal scoliosis); also, spinal column trauma (on one or more levels, what types of traumatic reconstruction? Spinal tumors - also what type (malignant, benign, etc.) and origin (bone, joint, connective, radicular)? Ossification of the posterior longitudinal ligaments at how many levels?; Spinal epidural hematoma (traumatic or spontaneous);
- Response: We appreciate your constructive comments. We have enhanced the manuscript with detailed descriptions of each condition, including specific diagnostic details and the extent of surgical interventions in subsection 2.1. Demographic Features of the Patients and Treatment Approach, page 3. This augmented information is now available in Table 1 and further detailed in the epidemiology subsection of the methodology section.
- It also remains unclear whether the infection was localized to the bony structures of the spinal vertebrae or whether it affected other elements as well (connective tissue, ligaments, spinal roots, deep muscles?; Wound debridement - it should be described in detail - did it affect only the skin and subcutaneous tissue or deeper elements (for example, muscle tissue and fascia).
- Response: As outlined in our paper's title, we focus on patients with postoperative deep wound infections. This encompasses infections affecting both the surrounding soft tissues and bony structures relevant to musculoskeletal and spinal surgery, in line with the widely accepted definition.
- -you did not mention anywhere in the paper the approval of the institution's Ethics committee for research
- Response: We acknowledge the omission and have amended the Institutional Review Board (IRB) approval statement (Page 14). The revised text reads: "Institutional Review Board Statement: This study was approved by the Institutional Review Board at the author’s institution (IRB No. 2023-06-017-002). As a retrospective clinical study, it received a waiver for informed consent."
II-2. Specific comments
1)-In all text of the article species of bacteriae were not written italics.
- Response: Following your advice, we have carefully revised the manuscript to ensure that the names of bacterial species are italicized throughout.
2) You have to check sentences in many parts they beginning with small letter and confusing reader-Also, you have to check all abbreviations to be explained when you firstly mentioned them
- Response We are grateful for your detailed review. The manuscript has been thoroughly revised for improved English expression and grammatical accuracy. Additionally, we have ensured consistent and clear use of abbreviations throughout the text.
3) Comments on the Quality of English Language
- Response: In line with your suggestion, extensive edits have been made to enhance the English language quality of the manuscript.
Additional revision in this manuscript we made :
Note: the supplementary revisions were made as follows beyond the comments and recommendations provided by the editor and reviewers.
- The affiliation (organization) has undergone minimal revisions; specifically, the term "convergence medicine" has been updated to "Biomedical science," and the number (1) has been added (page 1).
- The paper that was mistakenly labeled as a "Review" has been corrected to be categorized as an "Article." This error, along with an affiliation error, was brought to your attention on the day of submission via email.
- We revised Figure 1, adjusting the Y-axis from 1500 to 1000 micrograms/cm2 and adjusting the X-axis from 50 days to 42 days. Additionally, the range of days on the X-axis has been modified to include 7, 14, 21, 28, 35, and 42 days.
- In Figure 2, we added the daily release during the first week Figure 2 (a) followed by the overall release over 42 days (6 weeks) Figure 2 (b).
- We standardized the style and format of all figures (Figures 1, 2, and 3) in the manuscript to ensure uniformity.
- We included the data for vancomycin (VAN, 1.5 cm diameter) that was inadvertently omitted from the manuscript (pages 7 and 9).
- Sentences have been added on page 8, specifically within subsection 3.4.4 Cumulative Release Kinetics, offering insights into the dynamics of antibiotic release and the impact of bead size modification. Furthermore, on the same page and under the same subsection, the p-value, and AUC value have also been included.
- Table 1 has undergone minimal revisions (page 6), along with the addition of a footnote.
- The subsection under 2.2 titled "Preparation of PMMA DDS Beads" has been revised to "2.2. PMMA DDS Beads Formulation" on page 4.
- The contents of the subsection titled "2.4. Application of Mathematical Model for the estimation of release kinetics from Antibiotics-loaded PMMA Beads" have been segmented into four sub-headings: "2.4.1. Surface area considerations," "2.4.2. Mathematical Modeling in Drug Delivery Systems," and "2.4.3. Employment of the Higuchi Model." And “2.4.4. Bead Volume Calculation” on page 4.
- The subheadings "3.1. Clinical Outcome" and "3.3. Disease Entities" have been combined on page 5.
- The figure 1 caption has been revised (page 8).
- The footnote describing Figure 3 has been condensed (page 10.)
- The discussion section has been extended and subdivided into three sub-headings: 4.1. Antibiotic Release from PMMA Beads," "4.2. Impact of antibiotic levels in antibiotic-loaded bone cement (ALBC)," and "4.3. Combination Therapy in Chronic Osteomyelitis." Page 10~13).

Reviewer 2 Report
Comments and Suggestions for Authors
Madadi et al. showed the utility of mathematical modeling in drug release at deep surgical site infection. The author showed the impact of different physicochemical properties at the site of drug release. However, there are numerous limitations to the current study.
1) I don’t see the in vitro in vivo correlation (IVIVC) drug release profile. The author can show the estimated concentration of drug release and clinically observed concentration and see if it is within two-fold of the margin. Usually in Pharmacokinetics 2-fold criteria are well accepted.
2) This is a pure theoretical calculation-based study without showing any validation data for the calculated values, which makes the outcome of the results unwarranted. Hence, the author needs to generate solid in vitro drug release profile data and then validate it with the clinical outcome.
3) Currently, people are using more physiologically based pharmacokinetic (PBPK) modeling to understand the impact of different physicochemical properties on drug release profiles. Please, highlight the advantages of current mathematical modeling over PBPK modeling.
4) In the discussion section, the author highlighted the shortcomings of the current study rather than discussing current data and providing more literature evidence to support the data.
5) Please, define abbreviations on their first use.
6) Please, make the referencing style consistent.
Comments on the Quality of English Language
Need some grammar check.
Author Response
Response to Reviewer Comments
2. Response to Reviewer 2
Comments and Suggestions for Authors: Madadi et al. showed the utility of mathematical modeling in drug release at deep surgical site infection. The author showed the impact of different physicochemical properties at the site of drug release. However, there are numerous limitations to the current study.
1) I don’t see the in vitro in vivo correlation (IVIVC) drug release profile. The author can show the estimated concentration of drug release and clinically observed concentration and see if it is within two-fold of the margin. Usually in Pharmacokinetics 2-fold criteria are well accepted.
- Response: Thank you for your comment. This study is a retrospective cohort study focusing on patients treated with a local drug delivery system for deep surgical site infections. It was not within the scope of this study to correlate drug release profiles in vivo or in vitro. We concur with the necessity of validating pharmacokinetics through in vitro studies. The foundation of this study is based on the theoretical outcomes of direct local drug delivery from in vitro experiments. Our retrospective analysis evaluates long-term clinical experiences through a mathematical predictive model. We acknowledge the importance of experimental research for scientific validation and have noted this in our conclusion.
2) This is a pure theoretical calculation-based study without showing any validation data for the calculated values, which makes the outcome of the results unwarranted. Hence, the author needs to generate solid in vitro drug release profile data and then validate it with the clinical outcome.
- Author’s response: We acknowledge the significance of validating our mathematical modeling approach in predicting antibiotic release kinetics through in-vitro release experiments, as suggested by the reviewer. The validation process will be comprehensively addressed in an upcoming paper, and we appreciate the reviewer's insightful comment in this regard.
- It is noteworthy to mention that numerous published papers employing the same material and mathematical model have already undergone validation through in vitro release experiments.
- Our rationale for advocating the utilization of this established mathematical model is rooted in the commonality of materials and models employed across these studies. The existing body of literature, validated through in vitro release experiments, provides a robust foundation for promoting the adoption of this mathematical model by clinicians as a reliable predictor of drug release for individual patients. This is particularly relevant given the impracticality of conducting in-vitro experiments for individual patients as a personalized treatment strategy.
- We appreciate the reviewer's attention to this aspect of our work and emphasize our commitment to further elucidate the validation process in our forthcoming publication.
3) Currently, people are using more physiologically based pharmacokinetic (PBPK) modeling to understand the impact of different physicochemical properties on drug release profiles. Please, highlight the advantages of current mathematical modeling over PBPK modeling.
- Response: As mentioned in our manuscript, the advantages of our mathematical modeling include the ability to elucidate complex interactions affecting drug release, thereby evaluating the safety and effectiveness of localized drug therapy. These models are crucial for predicting and controlling drug release in bone infections or deep surgical site infections (DSSIs). They enable the customization of drug release profiles, facilitating accurate predictions of treatment outcomes.
4) In the discussion section, the author highlighted the shortcomings of the current study rather than discussing current data and providing more literature evidence to support the data.
- Response: We appreciate your observation. As suggested, we have expanded the discussion section to include a more thorough discussion of our study results, supported by relevant literature evidence (pages 10~13).
5) Please, define abbreviations on their first use.
- Response: We have revised the manuscript to ensure all abbreviations are correctly defined at their first use.
6) Please, make the referencing style consistent.
- Response: The referencing style has been revised and made consistent with the submission guidelines of the journal.
7) Comments on the Quality of the English Language Need some grammar checks.
- Response: In response to your comments, we have conducted a comprehensive grammar check and professional English language editing of the entire manuscript.
Additional revision in this manuscript we made :
Note: the supplementary revisions were made as follows beyond the comments and recommendations provided by the editor and reviewers.
- The affiliation (organization) has undergone minimal revisions; specifically, the term "convergence medicine" has been updated to "Biomedical science," and the number (1) has been added (page 1).
- The paper that was mistakenly labeled as a "Review" has been corrected to be categorized as an "Article." This error, along with an affiliation error, was brought to your attention on the day of submission via email.
- We revised Figure 1, adjusting the Y-axis from 1500 to 1000 micrograms/cm2 and adjusting the X-axis from 50 days to 42 days. Additionally, the range of days on the X-axis has been modified to include 7, 14, 21, 28, 35, and 42 days.
- In Figure 2, we added the daily release during the first week Figure 2 (a) followed by the overall release over 42 days (6 weeks) Figure 2 (b).
- We standardized the style and format of all figures (Figures 1, 2, and 3) in the manuscript to ensure uniformity.
- We included the data for vancomycin (VAN, 1.5 cm diameter) that was inadvertently omitted from the manuscript (pages 7 and 9).
- Sentences have been added on page 8, specifically within subsection 3.4.4 Cumulative Release Kinetics, offering insights into the dynamics of antibiotic release and the impact of bead size modification. Furthermore, on the same page and under the same subsection, the p-value, and AUC value have also been included.
- Table 1 has undergone minimal revisions (page 6), along with the addition of a footnote.
- The subsection under 2.2 titled "Preparation of PMMA DDS Beads" has been revised to "2.2. PMMA DDS Beads Formulation" on page 4.
- The contents of the subsection titled "2.4. Application of Mathematical Model for the estimation of release kinetics from Antibiotics-loaded PMMA Beads" have been segmented into four sub-headings: "2.4.1. Surface area considerations," "2.4.2. Mathematical Modeling in Drug Delivery Systems," and "2.4.3. Employment of the Higuchi Model." And “2.4.4. Bead Volume Calculation” on page 4.
- The subheadings "3.1. Clinical Outcome" and "3.3. Disease Entities" have been combined on page 5.
- The figure 1 caption has been revised (page 8).
- The footnote describing Figure 3 has been condensed (page 10.)
- The discussion section has been extended and subdivided into three sub-headings: 4.1. Antibiotic Release from PMMA Beads," "4.2. Impact of antibiotic levels in antibiotic-loaded bone cement (ALBC)," and "4.3. Combination Therapy in Chronic Osteomyelitis." Page 10~13).

Round 2
Reviewer 2 Report
Comments and Suggestions for Authors
The author addressed my comments satisfactorily.